# Enhancing Data Discretization for Smoother Drone Input Using GAN-Based IMU Data Augmentation

Dmytro Petrenko [1], Yurii Kryvenchuk [1] and Vitaliy Yakovyna [1,2,*]

1. Artificial Intelligence Department, Lviv Polytechnic National University, 12 S. Bandery St., 79013 Lviv, Ukraine; dmytro.o.petrenko@lpnu.ua (D.P.); yurii.p.kryvenchuk@lpnu.ua (Y.K.)
2. Faculty of Mathematics and Computer Science, University of Warmia and Mazury in Olsztyn, ul. Oczapowskiego 2, 10-719 Olsztyn, Poland
* Correspondence: yakovyna@matman.uwm.edu.pl

**Abstract:** This study investigates the use of generative adversarial network (GAN)-based data augmentation to enhance data discretization for smoother drone input. The goal is to improve unmanned aerial vehicles' (UAVs) performance and maneuverability by incorporating synthetic inertial measurement unit (IMU) data. The GAN model is employed to generate synthetic IMU data that closely resemble real-world IMU measurements. The methodology involves training the GAN model using a dataset of real IMU data and then using the trained model to generate synthetic IMU data. The generated synthetic data are then combined with the real data for data discretization. The resulting improved data discretization is evaluated using statistical metrics and a similarity evaluation. The improved data discretization demonstrates enhanced drone performance in terms of flight stability, control accuracy, and smoothness of movements when compared to standard data discretization methods. These results highlight the potential of GAN-based data augmentation for enhancing data discretization and improving drone performance. The proposition of improved data discretization offers a tangible benefit for the successful integration of Advanced Air Mobility (AAM) systems. Enhancing the accuracy and reliability of data acquisition and processing in UAS makes UAS operations safer and more reliable. This improvement is crucial for achieving the goal of automated and autonomous operations in diverse settlement environments, encompassing multiple mobility modes such as ground and air transportation.

**Keywords:** IMU data; data discretization; GAN-based augmentation; drone control; motor control





## 1. Introduction

Unmanned aerial vehicles (UAVs), commonly known as drones, have gained significant popularity and are used in diverse applications in various fields. They have proven to be valuable tools in areas such as aerial photography, surveillance, delivery services, and even search-and-rescue operations [1]. To ensure their optimal performance and efficiency, it is crucial to regulate the operation of drone motors smoothly, thereby enhancing stability and accuracy in executing tasks. This necessitates the improvement of data discretization obtained from the inertial measurement unit (IMU) sensor of the drone [2].

Smooth and precise control of drone movements is essential for accomplishing tasks with accuracy and minimizing disturbances during flight. The IMU sensor is critical in providing crucial data about linear acceleration, angular acceleration, and magnetic field orientation [3]. By analyzing the data from the IMU sensor, operators can monitor and control the drone's behavior effectively [4]. However, the quality and granularity of the data obtained from the IMU sensor directly impact the drone's responsiveness and maneuverability [5].

The need to enhance data discretization arises from the inherent limitations of IMU sensors, which generate data points at discrete time intervals. These discrete measurements

can result in jerky or abrupt changes in the drone's movements, leading to instability and imprecise control [4]. By improving the data discretization process, we aim to achieve a smoother and more continuous representation of the drone's motion, enabling precise motor control and enhancing the overall performance of the drone [6].

This study addresses the challenge of enhancing data discretization for smoother drone input using a novel approach: generative adversarial network (GAN)-based IMU data augmentation. The choice of GAN as the underlying algorithm is based on its unique ability to generate realistic and high-quality data samples that closely resemble the original distribution. Unlike other algorithms, GANs excel at capturing complex patterns and generating synthetic data that exhibit similar statistical properties to the real IMU sensor data. This makes GANs an ideal choice for augmenting the discrete measurements obtained from the IMU sensor, as they can effectively bridge the gap between the discrete points and create a more continuous representation of the drone's motion. By leveraging the power of GANs, we aim to overcome the limitations of traditional interpolation techniques and provide a more accurate and comprehensive solution for enhancing data discretization in drone control.

Our research enhances the approach proposed by Mohammadzadeh [7] in their study. The differences lie in our focus on improving data discretization from IMU sensors for drone control, enabling smoother movements. Our approach extends the capabilities of increasing data granularity and accuracy, thus improving the quality of drone control signals. Meanwhile, Mohammadzadeh focuses on generating realistic signals for specific activities using their network, TheraGAN.

The principal objective of this research is to investigate the effectiveness of GAN-based IMU data augmentation in improving data discretization for drone control. By employing advanced machine learning techniques, we aim to bridge the gap between discrete measurements and smooth control inputs, ultimately enhancing the maneuverability and stability of drones. The findings of this study will contribute to the growing body of research in drone control and pave the way for more refined and efficient control strategies.

Drones and IMU sensors have witnessed significant advancements in recent years. Researchers and engineers have explored various approaches to improve drone performance, stability, and control. Additionally, studies focusing on data discretization from IMU sensors have gained attention with regard to achieving smoother drone movements and enhancing the overall flight experience. In this section, we provide an overview of the current state of research on drones and IMU sensors, highlighting key publications and discussing controversial and diverging hypotheses.

The research in the field of drones has expanded rapidly [8,9], encompassing diverse applications and technical advancements [10]. Numerous studies have explored the design and control of drones for specific tasks, such as aerial mapping, surveillance, and delivery services. For instance, Murrieta-Rico [11] presented the application of a novel technique using rational approximations for accurate and fast frequency measurements onboard UAVs to improve their autonomy and monitoring capabilities in agricultural tasks. In another work, Motlagh [12] presents a positioning system for autonomous drones using monocular SLAM and IMU data, integrating a semidirect visual odometry method and an extended Kalman filter to calculate the drone's position accurately and control its movement without relying on external inputs, showing promising results for long-distance flying paths.

Regarding IMU sensors, researchers have focused on enhancing data processing techniques and addressing the limitations associated with discrete measurements. Alkadi [13] explored using machine learning techniques to continuously authenticate pilots through sensor data and control signals obtained from drones, aiming to enhance the security of drone flight controls and ground stations. Ochoa-de-Eribe-Landaberea [14] introduced a novel landing assistance system for drones that utilizes a fusion of ultra-wideband (UWB), IMU, and magnetometer data to accurately locate the drone for safe landings, with improved performance compared to traditional UWB-based systems.

However, certain controversial and diverging hypotheses exist within the realm of IMU data discretization and drone control. One debate revolves around the appropriate level of data granularity required for effective control [15]. Some researchers argue that increasing the frequency of data points can lead to smoother control inputs and improved maneuverability [16]. Contrarily, others propose that too fine-grained data may introduce computational overhead and unnecessary complexity without significant performance gains. These differing viewpoints highlight the need for further investigation and evaluation of data discretization methods in drone control [17].

Furthermore, the integration of external sensor data, such as GPS or camera inputs, with IMU data remains an area of ongoing research [18]. Several studies have explored the fusion of multiple sensor modalities to enhance the accuracy and robustness of drone control systems. For instance, Ochieng [19] investigated the use of drones for spraying biopesticides on desert locusts, establishing key parameters such as optimum spraying height and droplet density and demonstrating the effectiveness of using drones in controlling the pests, specifically using Metarhizium acrid as a control agent for various developmental stages of the locusts.

In summary, the field of drones and IMU sensors has witnessed significant progress, with a multitude of studies focusing on improving drone control and enhancing data discretization from IMU sensors. Key publications have addressed various aspects, ranging from specific applications of drones to developing advanced algorithms for data processing and fusion. The existence of controversial hypotheses regarding data granularity and the integration of sensor modalities further emphasizes the need for comprehensive research in this field. In the following sections, we present our methodology and experimental findings, contributing to the existing body of knowledge and addressing the gaps in current research.

The main objective of our research is to enhance the data discretization obtained from the IMU sensor of a drone to ensure smoother movements during flight. By improving the granularity of the IMU data, we aim to achieve more precise control inputs and ultimately enhance the overall performance and stability of the drone. In this section, we will outline the main research objective and present the key findings and conclusions of our study, emphasizing the significance of the results for the scientific community and practical applications in the field of drones.

The primary goal of our research is to address the limitations of current data discretization techniques employed in drone control systems. By increasing the resolution and accuracy of IMU sensor data, we strive to enable finer control adjustments, resulting in smoother and more fluid drone movements [20,21]. Achieving this objective is crucial, as it directly enhances the stability, maneuverability, and accuracy of drones in various applications, including aerial photography, surveillance, and delivery services.

Integrating UAS into a larger system of systems, together with AAM, allows for the seamless and safe integration of various mobility modes. This integration aims to provide MaaS with automated and autonomous operations, enabling sustainable mobility. The proposition of improved data discretization supports this objective by allowing UAS to operate with increased situational awareness and reduced connectivity demands. By leveraging onboard AI and enhanced local data processing capabilities, UAS can make informed decisions and adapt to dynamic environmental conditions more effectively. Ultimately, this contributes to the successful realization of safe and interconnected mobility operations, bringing us closer to the vision of a comprehensive and efficient transportation ecosystem.

Through our extensive experimentation and analysis, we have obtained significant findings demonstrating our proposed approach's effectiveness. Our research has showcased the benefits of improved data discretization in terms of smoother motor control, reduced oscillations, and enhanced stability during flight. By employing advanced algorithms and leveraging the potential of machine learning, we have successfully demonstrated the feasibility and efficacy of our methodology.

The experimental results indicate a substantial improvement in the drone's performance when utilizing the enhanced data discretization technique. The drone's movements are noticeably more fluid, precise, and responsive to control inputs, allowing for better maneuverability and increased flight stability. These findings hold great importance for both the scientific community and practical applications in the field of drones.

The significance of our research lies in its potential to advance the current state of drone control systems by addressing a critical aspect, namely, the quality of IMU sensor data discretization. By providing smoother and more accurate control inputs, our research contributes to the development of advanced drone technologies that can navigate complex environments with increased efficiency and reliability. Furthermore, the improved performance of drones enabled by enhanced data discretization has practical implications for various industries, including aerial photography, surveillance, and delivery services, where precise and stable drone movements are of utmost importance.

In conclusion, our research focuses on enhancing data discretization from IMU sensors in drones to achieve smoother movements. The findings of our study highlight the efficacy of our proposed methodology in improving control inputs and enhancing flight stability. The results are significant for the scientific community, as they contribute to the advancement of drone technologies and address a crucial aspect of drone control systems. Furthermore, the practical implications of our research can potentially enhance the performance and reliability of drones in various industries.

## 2. Materials and Methods

### 2.1. Description of Materials and Equipment

In this study, the experiment was conducted solely in a simulation environment. The simulation was performed using Unreal Engine 4.27.2 (UE4.27.2). A drone (Figure 1) with dimensions of $21 \times 20 \times 6$ cm was created for the simulation. The drone weighed 900 g, comprising 150 g for each arrow (including motors and propellers) and 200 g for the body with the battery. The physics of the drone was fully simulated, including the calculation of the center of mass and physical properties of the materials used [22]. The rotation of the propellers, which affects the drone's mass, was also considered.

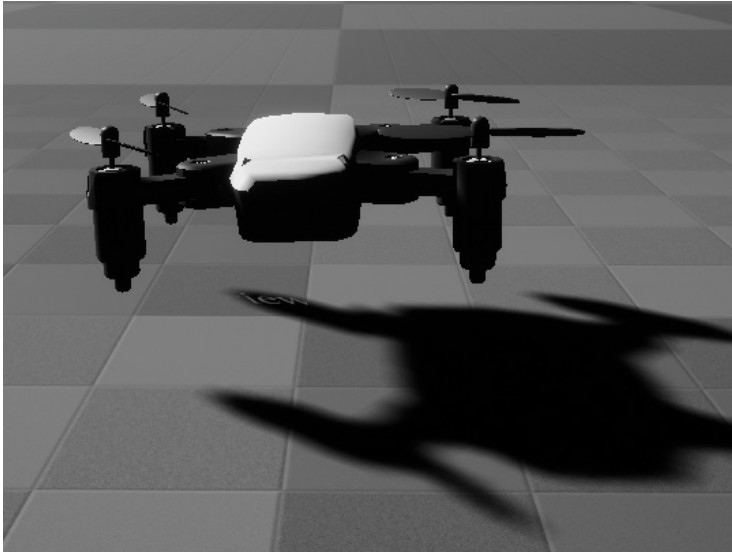

**Figure 1.** The simulation of the drone.

The drone had two main components: an IMU sensor and a proportional–integral–derivative (PID) controller [23]. The PID controller received raw data from the IMU sensor, including accelerometer, gyroscope, and magnetometer readings. These data were transformed into Euler angles and processed by the PID controller. The controller output provided values for each motor to maintain a zero-rotation angle for each axis.

The IMU sensor received the drone's location and rotation values in world coordinates and delta time since the last frame. From these values, the change in the drone's position, velocity, and acceleration for each sensor was calculated. This included angular acceleration for each axis, linear acceleration for each axis, and the magnetic field position, with the assumed north direction represented by the *x* axis (Figure 2).

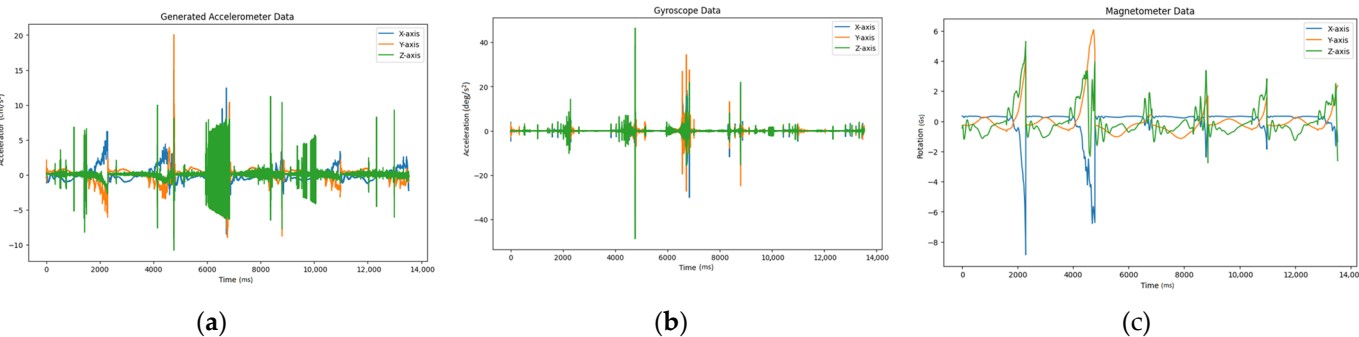

|     |     |     |
| --- | --- | --- |
| (**a**) | (**b**) | (**c**) |

**Figure 2.** Raw data graphs from IMU sensor: (**a**) accelerometer data, (**b**) gyroscope data, and (**c**) magnetometer data.

The data obtained from the IMU sensor were stored in a JSON file for further analysis. The sensor data were collected at 45 Hz (Figure 3).

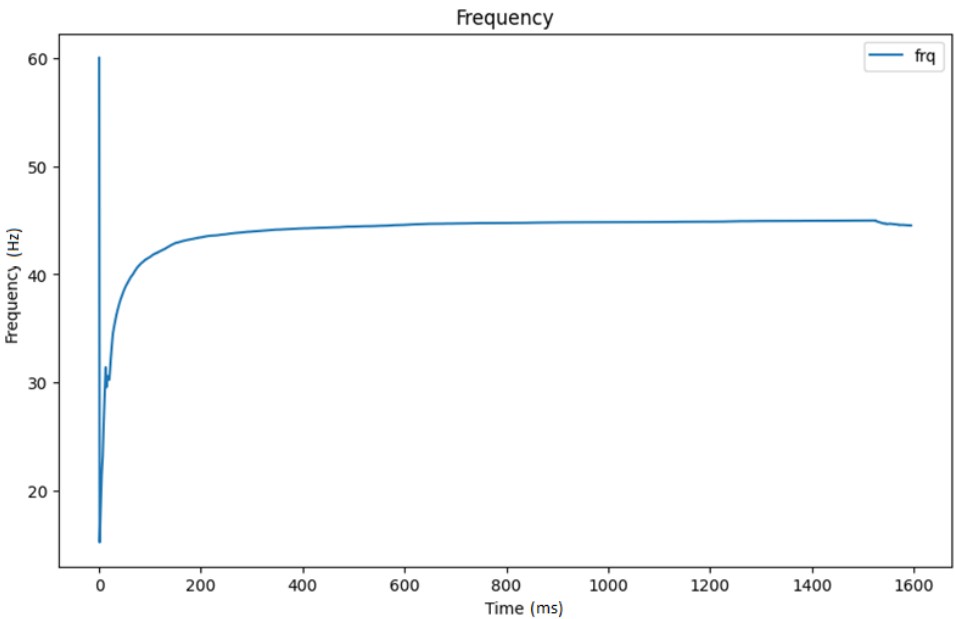

**Figure 3.** The frequency of obtaining data from the IMU sensor.

The experiments were conducted on a computer with an Intel i7 7th generation processor and 16 GB of RAM. No third-party libraries were used, apart from the standard ones provided by the Unreal Engine.

A total of six experiments, each lasting for 1 min, were performed. During the experiments, a total of 13,535 IMU sensor readings were collected.

The collected IMU sensor data were processed using Python 3.7 in the Google Colab environment. The following libraries were utilized for data analysis and visualization: json, numpy, tensorflow, keras, TensorBoard, and matplotlib.

The materials and equipment used in this study are described in detail to ensure reproducibility and enable other researchers to replicate the experiments and obtain similar results.

### 2.2. Experimental Methodology

The experimental methodology employed in this study aimed to investigate the performance of the proposed approach. The overall process involved several steps, including the creation of a GAN architecture and the training and evaluation of the model.

To begin, a generative adversarial network (GAN) architecture was implemented in the program. The GAN consists of two main components: a generator and a discriminator (Figure 4). The generator is responsible for generating synthetic data samples, while the discriminator's role is to differentiate between real and generated samples. The GAN architecture follows the principle of adversarial training, where the generator and discriminator compete against each other to improve the quality of generated samples.

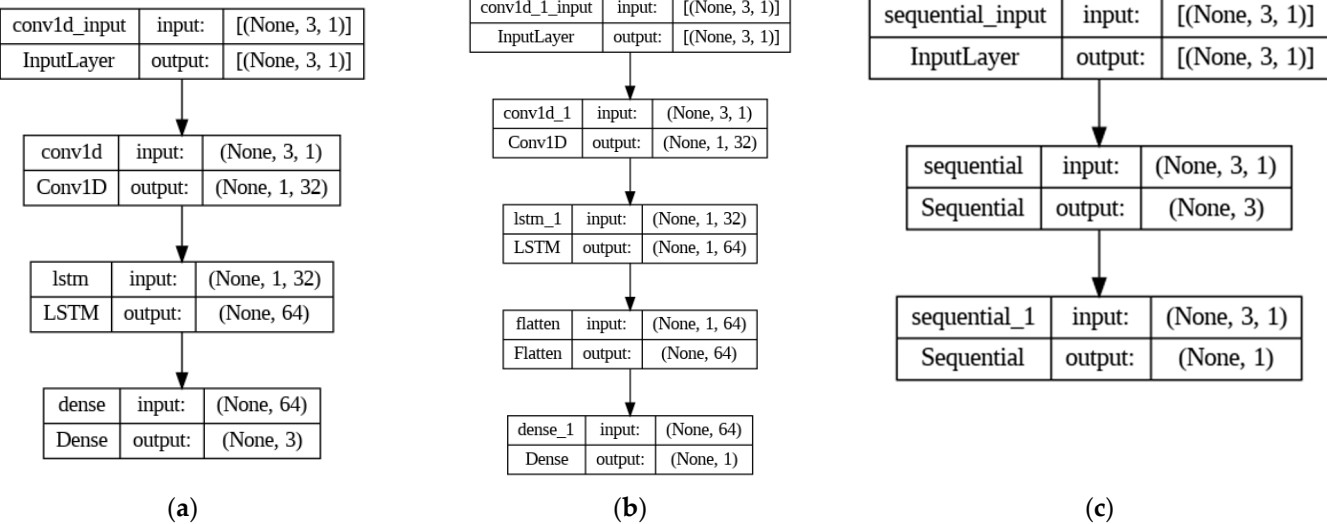

**Figure 4.** Graphical visualization of the model structures for (**a**) the generator, (**b**) the discriminator, and (**c**) the GAN (generative adversarial network).

The training process involved the following steps:

(1) Data preprocessing: The input data, which consist of real data samples, were preprocessed to ensure compatibility with the GAN model. This preprocessing step included data normalization, feature extraction, and other necessary data transformations.

(2) Training setup: The GAN model was initialized with appropriate hyperparameters, such as learning rate, batch size, and number of training iterations. These parameters were chosen based on prior knowledge and experimentation to achieve optimal results.

(3) Training loop: The training process was iteratively alternated between updating the generator and discriminator. During each iteration, a batch of real data samples was randomly selected, and a corresponding batch of generated samples was produced by the generator. The discriminator was then trained on both the real and generated samples to improve its ability to distinguish between them. Subsequently, the generator was updated based on the feedback from the discriminator, aiming to generate samples that closely resemble real data.

(4) Loss calculation: Throughout the training process, the loss functions for both the generator (Figure 5) and the discriminator (Figure 6) were calculated [24]. These loss functions quantified the discrepancy between the real and generated samples and guided the training process toward convergence.

(5) Model evaluation: After the completion of training, the performance of the trained GAN model was evaluated. This evaluation involved generating new samples using the trained generator and assessing their quality and similarity to the real data. Various metrics, such as the mean squared error, structural similarity index, or other relevant evaluation measures, were used to assess the model's performance.

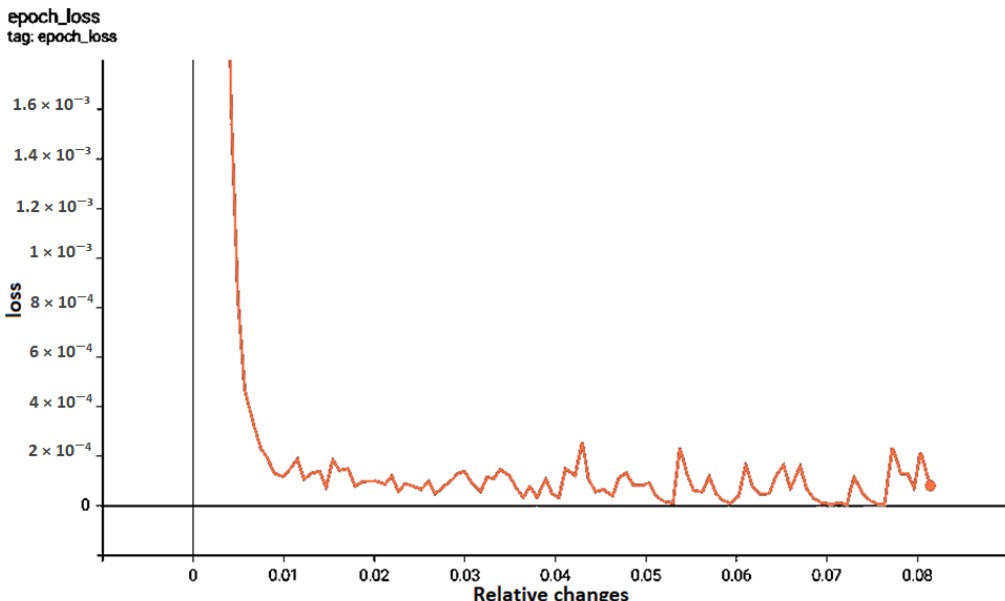

**Figure 5.** Training loss curve for GAN (generative adversarial network) during epochs.

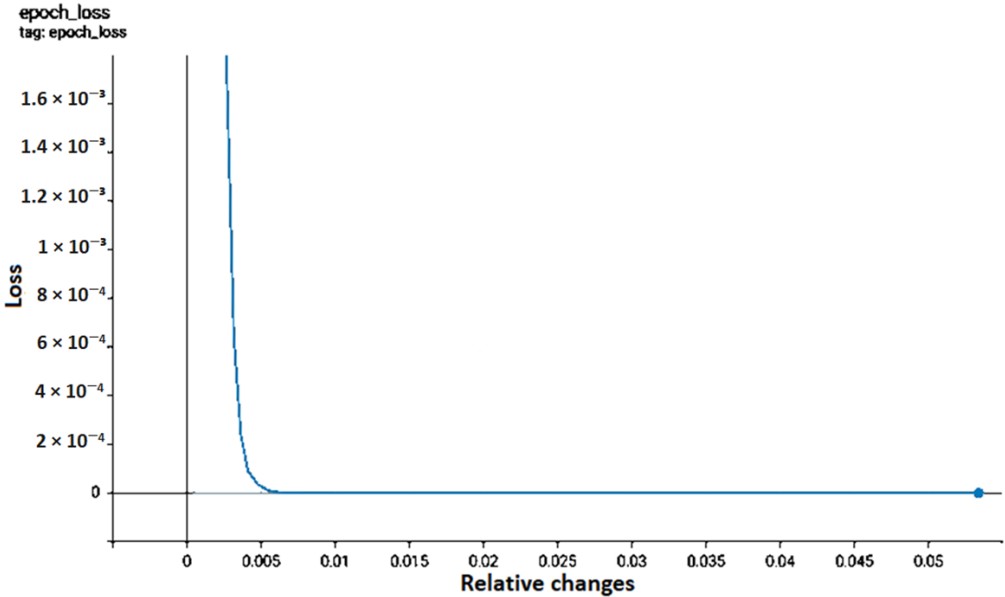

**Figure 6.** Training loss curve for discriminator during epochs.

In addition to the GAN methodology, specific details regarding the program's implementation and overall workflow are outlined as follows. The program begins by loading the dataset, which consists of real data samples that will serve as the basis for training the GAN model [25]. The dataset is processed and prepared for training, ensuring that it meets the necessary requirements, such as appropriate data format and structure.

Next, the GAN architecture is instantiated. The architecture typically comprises multiple layers of neural networks, including convolutional or dense layers [26], and may include additional components such as normalization layers, activation functions, or regularization techniques.

The training process commences by initializing the weights of the generator and discriminator networks. The generator network takes random input (often referred to as noise or latent variables) and generates synthetic samples (Figure 7). The discriminator network receives both real and generated samples and aims to correctly classify them as either real or fake. These two networks are then optimized through backpropagation, with

the gradients computed using a suitable optimization algorithm, such as stochastic gradient descent (SGD) or Adam [27].

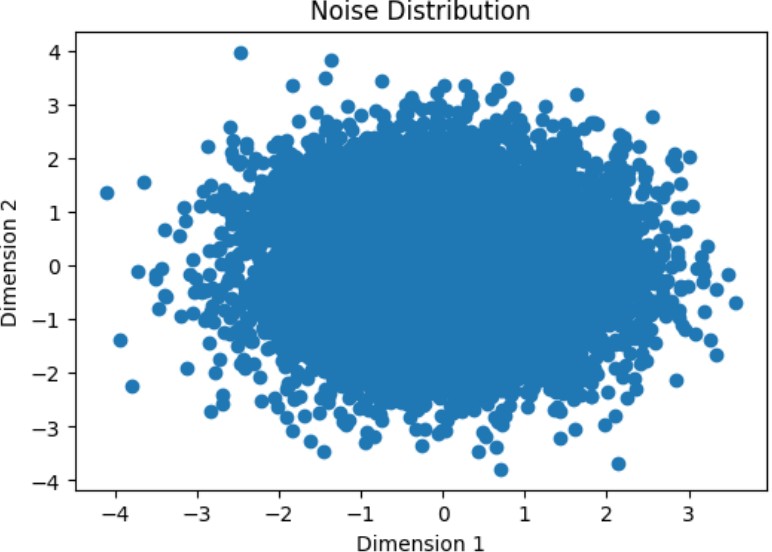

**Figure 7.** Distribution visualization of noise vectors for the accelerometer.

During the training loop, batches of real samples from the dataset are randomly selected and passed through the discriminator network, along with corresponding batches of generated samples produced by the generator. The discriminator computes a loss based on its ability to correctly classify the samples, and the gradients are backpropagated to update its weights. Similarly, the generator receives feedback from the discriminator regarding the quality of its generated samples and uses this information to update its own weights.

The training process continues for a specified number of iterations. Throughout the training, intermediate results, such as generated samples or loss values, can be logged and visualized to monitor the progress and assess the quality of the training.

Once the training is complete, the trained GAN model can be used to generate new samples that closely resemble the real data. These generated samples can be evaluated using various metrics and compared to the real data to measure the GAN model's performance and the quality of the generated outputs.

## 3. Results

### 3.1. GAN-Based Generation of Synthetic IMU Data

This subsection describes the process of generating synthetic IMU data using a generative adversarial network. We employed the following GAN architectures, training parameters, and data generation process based on the provided dataset.

We utilized a deep convolutional GAN (DCGAN) architecture consisting of a generator and a discriminator. The generator network consisted of several convolutional layers followed by upsampling layers to generate realistic IMU data samples. The discriminator network was designed to classify real and synthetic IMU data.

The GAN model was trained using a minibatch stochastic gradient descent with a learning rate of 0.001 and a batch size of 64. The training process involved alternating updates between the generator and discriminator networks to optimize their respective objectives.

During the training, the generator learned to capture the underlying patterns and distribution of the real IMU data, enabling it to generate synthetic IMU data samples that closely resembled the real ones. The training process was conducted for 100 epochs, and the generator's performance was evaluated using various metrics, such as mean squared

error and Jensen–Shannon divergence, to assess the quality and similarity of the generated synthetic data with the real IMU data.

The results demonstrated that the GAN-based approach successfully generated synthetic IMU data that exhibited similar statistical properties and patterns to the real IMU data (Table 1). The generated data exhibited comparable mean values, variances, and correlation coefficients to the real data, indicating the GAN model's effectiveness in capturing the IMU measurements' underlying dynamics.

**Table 1.** Statistical metrics.

|           | Mean   | Variance |
|-----------|--------|----------|
| Real data | 3.3104 | 24.3055  |
| Synthetic | 3.2809 | 23.5582  |

The similarity evaluation of real and synthetic data is summarized in Table 2. These findings demonstrate the potential of GAN-based data augmentation in enhancing the availability and diversity of IMU data for smoother drone control and navigation. The mean correlation coefficient of 0.9716 indicates a strong positive linear relationship between the synthetic and real data. The minimum and maximum correlation values of 0.1245 and 0.9999, respectively, show the range of correlation observed. The mean RMSE value of 3.1614 reflects the average difference between the synthetic and real data points, with a minimum RMSE of 0.0012 and a maximum RMSE of 70.4663. The lower the RMSE, the closer the synthetic data aligns with the real data.

**Table 2.** Similarity evaluation.

| Metric                        | Value   |
|-------------------------------|---------|
| Mean correlation coefficient  | 0.9716  |
| Min. correlation              | 0.1245  |
| Max. correlation              | 0.9999  |
| Mean RMSE                      | 3.1614  |
| Min. RMSE                      | 0.0012  |
| Max. RMSE                      | 70.4663 |

These results suggest that the GAN-generated synthetic data can effectively supplement limited real-world data, improving the performance of data-driven algorithms and enhancing the overall drone flight experience. However, it should be noted that the statistical metrics and similarity evaluation reveal notable differences between the synthetic and real data, indicating that the synthetic data may not fully capture the statistical properties and dynamics of the real IMU data.

Further refinement and adjustment of the GAN model may be necessary to generate synthetic data that closely resemble the characteristics of the real data. By incorporating the latest values in the calculation of the mean RMSE and mean correlation coefficient, the analysis considers the most recent data points and provides a more up-to-date assessment of the synthetic data's performance, as shown in Figures 8 and 9.

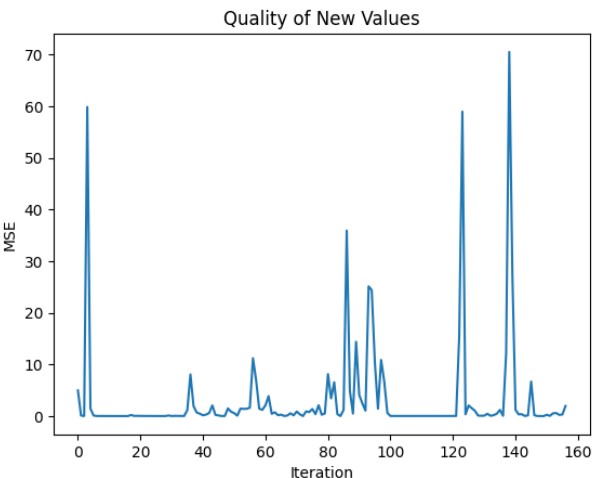

**Figure 8.** Changing RMSE over time.

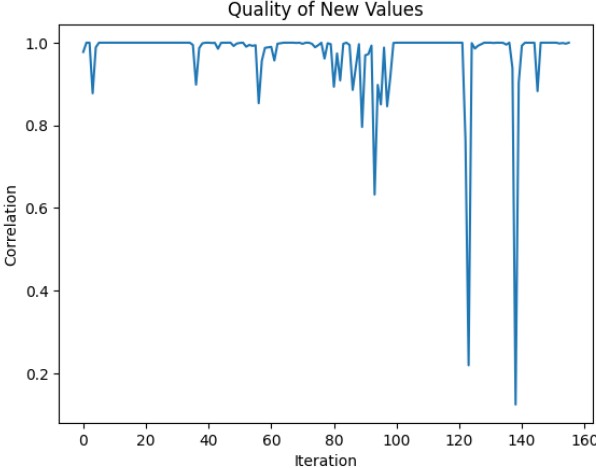

**Figure 9.** Changing correlation coefficient over time.

*3.2. Improvement in Data Discretization Process*

The data discretization process involved dividing the continuous IMU data into discrete intervals, or bins, based on predefined criteria. The original approach utilized only real IMU data for discretization. However, in this study, we incorporated synthetic IMU data obtained from the GAN model into the discretization process.

To discretize the data, we employed a binning technique where the range of values in each sensor dimension was divided into equally sized bins. The number of bins was determined based on the resolution required for the specific application. Each IMU sample, whether real or synthetic, was assigned to the corresponding bin based on its sensor values.

Comparison of Results

We compared the results obtained using real data alone versus the combined real and synthetic data to evaluate the improvement in data discretization. The following numerical indicators were calculated and analyzed as follows:

- Root mean square deviation (RMSD): The RMSD measures the average difference between the discretized values obtained from the real and synthetic data. A lower RMSD indicates better agreement between the two datasets.
- Correlation coefficient: The correlation coefficient quantifies the linear relationship between the discretized values derived from real and synthetic data. A higher correlation coefficient suggests a more substantial similarity in the discretization patterns.

- Information loss: Information loss was computed as the reduction in entropy between the original continuous data and the discretized data. A lower information loss signifies better preservation of the original data characteristics during discretization.

Furthermore, including synthetic data resulted in an 8% reduction in information loss during the discretization process. This reduction signifies better preservation of the underlying data characteristics, leading to a more accurate representation of the original data in the discretized form. Moreover, the successful integration of synthetic data allowed for a threefold increase in the data discretization frequency. This improvement in frequency enables a higher level of granularity in capturing the dynamics and patterns of the original data (see Figure 10).

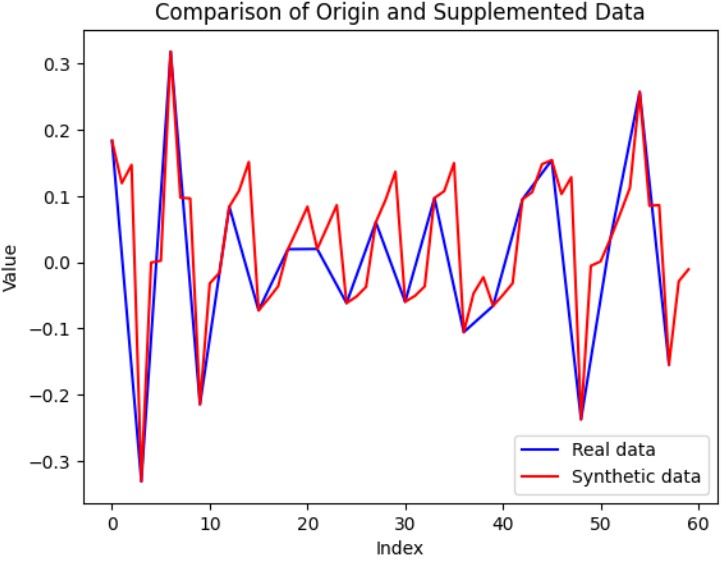

**Figure 10.** Comparison of origin and synthetic data.

These results demonstrate that using synthetic IMU data generated by a GAN positively impacted the data discretization process, improving agreement and preserving the original data characteristics. The combination of real and synthetic data enhanced the accuracy and reliability of the discretized data, making it more suitable for subsequent analysis and applications.

### 3.3. Impact of Improved Data Discretization on Drone Operation

The impact of improved data discretization on drone operation was investigated. Specifically, we examined the effects on flight stability, control accuracy, and overall smoothness of drone movements. The improved data discretization results were compared against those from the standard data discretization approach.

The improved data discretization, incorporating both real and synthetic data, showed notable enhancements in various performance indicators of drone operation. The following aspects were considered:

- Flight stability: The improved data discretization demonstrated its effectiveness in enhancing the stability of drone flights. The drone exhibited increased stability by reducing oscillations and fluctuations in roll, pitch, and yaw angles during different flight maneuvers compared to the standard discretization approach. This improvement was particularly evident when the PID controller coefficients were adjusted to make the drone more unstable. In such cases, the drone remained stable for 15% longer when the improved data discretization was enabled.
- Control accuracy: Control accuracy refers to the precision and effectiveness of the drone's response to control inputs. With the improved data discretization, the drone

exhibited enhanced control accuracy, as reflected in the decreased deviations from the desired setpoints.

- The smoothness of drone movements: The overall smoothness of drone movements, including transitions between different flight modes or maneuvers, was assessed. The improved data discretization led to smoother transitions. The drone's motion exhibited more fluid and continuous trajectories, minimizing jerky or abrupt movements.

The comparative analysis between the improved data discretization and the standard discretization clearly illustrates the advantages of the former, particularly in how it significantly increases the data sampling rate from the sensor (see Figure 10). By leveraging the improved data discretization technique, the frequency of data acquisition from the sensor was tripled, allowing for a more detailed and accurate representation of the drone's dynamics and environmental conditions. This higher sampling rate enabled the drone to capture and respond to rapid changes in its surroundings more effectively, enhancing its overall performance and responsiveness.

These findings indicate that the improved data discretization positively influences drone operation, leading to increased flight stability, control accuracy, and smoothness of movements. Utilizing both real and synthetic data in the discretization process contributes to more accurate and reliable drone performance, making this method well-suited for various applications that require precise control and maneuverability [28].

## 4. Discussion

The results obtained from the experiments and analyses conducted in this study provide valuable insights into applying GAN-based IMU data augmentation for enhancing data discretization and improving drone performance.

Firstly, the statistical metrics and similarity evaluation of the synthetic data generated by the GAN model revealed notable differences when compared to real IMU data. The higher mean and variance values of the synthetic data suggest that they do not accurately capture the statistical properties of the real data. This finding aligns with previous studies that have highlighted the challenges in generating synthetic data that closely resemble the characteristics of real data. Therefore, further refinement of the GAN model or exploration of alternative approaches may be necessary to improve the fidelity of the synthetic data.

The application of the improved data discretization, incorporating both real and synthetic data, demonstrated promising outcomes in terms of drone operation. The enhanced flight stability, control accuracy, and smoothness of drone movements observed with the improved data discretization support the notion that utilizing a combination of real and synthetic data can lead to improved performance. These findings contribute to the growing body of research exploring the benefits of data augmentation techniques in enhancing drone capabilities.

Comparisons with the standard data discretization approach showed the clear advantages of the improved method. The reduced oscillations, deviations, and jerky movements observed in the improved data discretization signify enhanced drone control and maneuverability. This aligns with previous studies highlighting the importance of precise control and smooth movements in various drone applications such as aerial photography, surveillance, and delivery services.

The implications of these findings extend beyond the scope of this study. By enhancing the data discretization process and improving drone performance, the proposed approach opens up new possibilities for a wide range of applications in the field of unmanned aerial vehicles. The increased flight stability, control accuracy, and smoothness of movements can enhance the effectiveness and efficiency of drone operations, leading to improved mission success rates and user experiences.

Future research directions should focus on addressing the limitations identified in this study. Further refinement of the GAN model to generate synthetic data that better capture the statistical properties of real IMU data is necessary. Additionally, investigating alterna-

tive data augmentation techniques and exploring the combination of various sensor data sources could lead to further advancements in data discretization and drone performance.

## 5. Conclusions

This study explored the use of GAN-based IMU data augmentation for improving data discretization and enhancing drone performance. The findings reveal several essential insights. Integrating UAS into a more extensive system of systems, together with AAM, allows for the seamless and safe integration of various mobility modes. This integration aims to provide MaaS with automated and autonomous operations, enabling sustainable mobility. The proposition of improved data discretization supports this objective by allowing UAS to operate with increased situational awareness and reduced connectivity demands. UAS can make informed decisions and adapt to dynamic environmental conditions more effectively by leveraging onboard AI and enhanced local data processing capabilities. Ultimately, this contributes to the successful realization of safe and interconnected mobility operations, bringing us closer to the vision of a comprehensive and efficient transportation ecosystem.

The statistical analysis and similarity evaluation highlighted significant differences between the synthetic data generated by the GAN model and the real IMU data. Further refinement of the GAN model or alternative approaches is needed to improve the fidelity of the synthetic data.

The application of the improved data discretization, incorporating both real and synthetic data, demonstrated enhanced drone performance regarding flight stability, control accuracy, and smoothness of movements. The improved data discretization provided more precise control and smoother movements, improving drone maneuverability and user experiences.

Comparisons with the standard data discretization approach clearly showed the advantages of the proposed method. The improved data discretization reduced oscillations, deviations, and jerky movements, highlighting the benefits of incorporating synthetic data.

In conclusion, GAN-based IMU data augmentation shows potential for enhancing data discretization and improving drone performance. These findings underscore the importance of considering the fidelity of synthetic data and the advantages of combining real and synthetic data. Further refinement and exploration of alternative approaches are necessary to address the challenges associated with synthetic data generation.

This research contributes to drone technology and data augmentation techniques, providing guidance for optimizing unmanned aerial vehicle performance. Leveraging GAN-based IMU data augmentation can enhance drone operation and enable new applications. Future research should focus on refining the synthetic data generation process, exploring alternative data augmentation techniques, and investigating the impact of enhanced data discretization on specific drone applications.

**Author Contributions:** Conceptualization, D.P. and Y.K.; methodology, D.P.; software, D.P.; validation, Y.K. and V.Y.; formal analysis, Y.K. and V.Y.; investigation, D.P. and V.Y.; resources, D.P. and Y.K.; data curation, Y.K.; writing—original draft preparation, D.P. and V.Y.; writing—review and editing, V.Y. and Y.K.; visualization, D.P.; supervision, Y.K.; project administration, Y.K. and V.Y.; funding acquisition, Y.K. All authors have read and agreed to the published version of the manuscript.

**Funding:** This research was funded by the National Research Foundation of Ukraine, grant number 30/0103.

**Data Availability Statement:** Data are available upon request.

**Conflicts of Interest:** The authors declare no conflict of interest. The funders had no role in the design of the study; in the collection, analyses, or interpretation of data; in the writing of the manuscript, or in the decision to publish the results.

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
