# Peer review of "Enhancing Data Discretization for Smoother Drone Input Using GAN-Based IMU Data Augmentation"

_drones, doi:10.3390/drones7070463_

Round 1

Reviewer 1 Report

The authors use Generative Adversarial Network (GAN)-based data augmentation to enhance data discretization for smoother drone input to improve the performance and maneuverability of unmanned aerial vehicles by incorporating synthetic Inertial Measurement Unit (IMU) data. Overall, the entire study is scientifically sound. However, There are substantial issues needing to be addressed, and the major revision is suggested before publication .

First of all, it is not clear why the selected set of techniques(GAN) fits best to the tasks to enhance the data discretization obtained from the IMU sensor of a drone. In addition,  the experiment was conducted solely in a simulation environment. The simulation was performed using Unreal Engine 4.27.2. The difference between simulated and real data needs to be quantitatively evaluated, and the accuracy of the simulation is questionable,More experiments are needed.Moreover, the state-of-the-art is not complete (missing GAN data enhancement in the related filed ).

Further comments / questions:

AAM should be full writed at first time in the abstract.

There are no suggestions for English writing。

Author Response

Dear Reviewer, thank you for the detailed review of our paper and valuable comments and suggestions. We revised the paper considering the suggestions of all reviewers. A more detailed explanation is below.

1) First of all, it is not clear why the selected set of techniques(GAN) fits best to the tasks to enhance the data discretization obtained from the IMU sensor of a drone.

We added the motivation of using GAN to enhance data discretization in the fourth paragraph of the introduction section.

2) In addition,  the experiment was conducted solely in a simulation environment. The simulation was performed using Unreal Engine 4.27.2. The difference between simulated and real data needs to be quantitatively evaluated, and the accuracy of the simulation is questionable, More experiments are needed.

Thank you for your comment. Yes, this paper is a simulation study, as many others and the experimental evaluation and verification will be the goal of future research. However, here we compare in detail the simulated and real data, including statistical description. Such a description is listed in Tables 1 and 2, as well as in Fig. 10. Comparisons with the standard data discretization approach showed the advantages of the proposed method. The improved data discretization allows for reducing oscillations, deviations, and jerky movements, highlighting the benefits of incorporating synthetic data. The practical experiments with real drones’ IMU are beyond the scope of this study, but the verification of the obtained simulations, along with their improvement, is the topic of our future study, as described in the discussion and concluding sections.

3) Moreover, the state-of-the-art is not complete (missing GAN data enhancement in the related filed ).

We added the related works n GAN data enhancement and described the differences between our approach compared to the published studies.

4) AAM should be full writed at first time in the abstract.

Thank you, we have updated this issue in the abstract.

Reviewer 2 Report

I attach a full report

Author Response

Dear reviewer, thank you so much for the detailed and thorough review of our paper. It was a great pleasure to read the review, and we highly appreciate your time spent reading the paper and writing such a comprehensive review. We are aware of the weaknesses of the research presented in this paper and are grateful for valuable comments and suggestions. As you suggested, we consider investigating alternative data augmentation techniques and exploring the combination of various sensor data sources. As you mentioned, we will address those topics in our future research.

Reviewer 3 Report

The authors presented work to provide data augmentation for smoother drone input using GAN. The work is useful to provide alternative solution to deal with the problem of data discretization from IMU sensors in this area. The introduction is informative. The authors introduced the limitation of existing work, and also the motivation of their research. The methods are introduced with details. Minor comments: image quality can be further improved. The font size could be enlarged. It would also be better to change x-axis to standard unit, for example, seconds / minutes / hours. Some units are missing, for example, Hertz for frequency. It would be better to show Fig. 5 and 6 with white background.

Overall writing is fine. The authors may consider the format when they cite a reference. Normally only the last names of the authors from the reference are needed, full names are not necessary. 

Author Response

1) Minor comments: image quality can be further improved. The font size could be enlarged. It would also be better to change x-axis to standard unit, for example, seconds / minutes / hours. Some units are missing, for example, Hertz for frequency. It would be better to show Fig. 5 and 6 with white background.

Thank you very much for your valuable comments and suggestions. We updated figures in the paper: we added units to axes in Fig. 2 and 3; we added Hertz in Fig. 3 and replaced Fig. 5 and 6 with ones having a white background.

2) The authors may consider the format when they cite a reference. Normally only the last names of the authors from the reference are needed, full names are not necessary.

Thank you for your comment. We removed the first names of the cited authors and left only the last names.